# Prevalence and Risk Factors Associated with Rib Lesions in Dairy Cows

**DOI:** 10.3390/ani14020338

**Published:** 2024-01-22

**Authors:** George Stilwell, João M. B. Azevedo, Telmo Nunes

**Affiliations:** 1Animal Behaviour and Welfare Research Laboratory, Centre for Interdisciplinary Research in Animal Health, Veterinary Medicine Faculty, Lisbon University, Av. Universidade Técnica, Alto da Ajuda, 1330-477 Lisboa, Portugal; 2Associate Laboratory for Animal and Veterinary Sciences (AL4AnimalS), Faculty of Veterinary Medicine, University of Lisbon, 1300-477 Lisboa, Portugal

**Keywords:** dairy cow, rib lesions, welfare, pain, lameness, stall dimensions

## Abstract

**Simple Summary:**

Some dairy cows show uni or bilateral hard protuberances over certain ribs. Although not painful, these injuries may have welfare and economic implications. The cause for these lesions is not completely clear. We examined all 1319 lactating cows from 22 dairy farms in Portugal. In each farm, housing and animal potential risk factors were recorded. Additionally, clinical examination, including lameness scoring, was performed on all cows showing rib lesions. The global rib lesions’ prevalence per farm was 2.3%, ranging from 0% (n = 12 farms) to 6.1% (n = 1 farm). The main farm risk factors found were narrow stalls; insufficient feeding places and cubicles; and the presence of an exposed curb at the back of the stalls. Lesions were mainly found in older cows (average of 3.7 lactations). Rib lesions were also associated with a clinical history of lameness but not with ongoing lameness. Because rib lesions seem to be associated with chronic lameness and badly designed stalls, they could be used as an indicator in on-farm welfare assessment.

**Abstract:**

Unilateral or bilateral hard bony enlargement is sometimes palpated over dairy cows’ ribs. Although usually not painful, these injuries may have welfare and economic implications and can be used as indicators of poor husbandry conditions. The objective of this study was to determine the prevalence of rib lesions in dairy cows kept in intensive systems and to identify the risk factors either associated with the housing or with the animal’s clinical history. We examined the ribs of all 1319 lactating cows from 22 Portuguese dairy farms. In each farm, housing and animal potential risk factors were recorded. Additionally, clinical examination, including lameness scoring, was performed on all cows showing rib lesions. The global rib lesions’ prevalence per farm was 2.3%, ranging from 0% (n = 12 farms) to 6.1% (n = 1 farm). Lesions were found in cows with an average of 3.7 lactations. The main farm risk factors were narrow stalls; insufficient feeding places; insufficient cubicles; and the presence of an exposed curb at the back end of the stalls. Rib lesions were associated with a history of lameness but not with lame cows at the moment of examination or a history of being a downer cow. In conclusion, rib lesions’ prevalence can be high in some farms, being significantly associated with cubicle design and lameness. By being associated with chronic lameness and inadequate housing, rib lesions should be included in dairy cows’ welfare assessment protocols.

## 1. Introduction

Stall design plays a major role in the prevalence of many skin and muscular-skeletal lesions in dairy cows [1,2,3] and this impact may be more severe if lying down is not easy, as it happens with lame animals. These lesions have been described in various parts of the limbs (e.g., carpus, hock, and neck) but very rarely have ribs been included.

The bovine thoracic cavity is protected by 13 ribs on each side [4]. Some areas of these bones are easily exposed to trauma, which can be self-inflicted, caused by other animals, or by collisions with housing structures (stall dividing rails, for example) [1,2,3]. The location of the lesions may depend on the cause, meaning that various risk factors often come into play, and, consequently, different farms may have their own more prevalent injury location. The costochondral joint seems to be particularly fragile due to its instability and higher exposure to impacts when the animal lies down [5], while the dorsal part of the more caudal ribs is more exposed to collisions with stall structures. 

Although ribs are not usually carefully inspected during routine clinical examination, there are reports and references of hard and non-painful enlargements on different ribs of adult cows, especially at the costochondral junction of the eighth and ninth rib, usually just caudal to the olecranon [5]. Most cases are reported to be bilateral and symmetrical and occur in low-body-condition cows. 

During the first author’s clinical work as a practitioner, many of these lesions were seen and palpated, but their impact on cow welfare and performance has never been investigated until now. These lesions should not be mistaken with real fractured ribs that have been described in dairy cows, as these were limited to the caudal ones—the 13th rib in three animals and the 11th rib in one animal—and were associated with other morbidities (e.g., peritonitis) [6]. They should also be differentiated from lesions resulting from rib fractures pertaining to difficult calving [7], from falls on a slippery floor (e.g., when mounting cows in heat) or from head-butts during fighting. Other causes may be crashes against structures during loading, unloading, and transport [8], during handling in races and chutes [9], or due to poor lairage conditions [10].

The source and nature of these hard non-painful rib lesions are still unclear. They may correspond to rib fractures or exostoses caused by mild but frequent trauma. Radiographs and necropsy material [11,12,13] show that at least some of these enlargements over the costochondral junction correspond to bone callous. It is also still debatable if the non-painful growths are the chronic stage of an acute lesion when painful swelling goes unnoticed. It is suggested that because healing of the cartilaginous tissues is much slower lesions will have this chronic appearance [5]. 

Anecdotal evidence shows a close relationship between this type of rib lesions and lameness [5,9,13] especially in animals using inadequate free or tie stalls. This association could be explained by the fact that a lame cow will collide more easily with the stall structures or hard dividers or will crash against the floor or its own front hooves because of the difficulty in lying in a slow and controlled manner [3,5,14,15].

In Europe, the Welfare Quality project [16] includes lameness, collisions with stall structures, and swellings in their list of animal-based indicators used to assess welfare in dairy farms. To validate the inclusion of rib lesions in such welfare assessment protocols, it would be important to provide scientific evidence of the prevalence and the role played by different risk factors, including those used for assessment. Blowey [5] also suggests that these lesions could be a useful indicator in the assessment of disputed welfare cases related to lameness.

This paper aims to investigate the prevalence of this type of rib lesions in Portuguese dairy farms and to identify farm and animal risk factors. To our knowledge, this is the first scientific study on the prevalence of rib lesions in different housing systems to be published.

## 2. Materials and Methods

Ethical review and approval were waived for this study because it only included palpation of the coastal region of animals restrained for routine and mandatory sanitary interventions. It was demonstrated to the Committee that the study did not involve any experimental procedure or specific handling of cattle that could cause any level of distress, pain, or suffering. A declaration from the Ethics Committee of the Faculty of Veterinary Medicine, University of Lisbon, was issued and is available for consultation.

### 2.1. Farm and Animal Recruitment

The study was carried out in the south and center of Portugal where small and medium size family-run farms are the basis of the dairy industry. Basic information about each farm was gathered prior to the first visit. This included herd size, average milk yield, and housing system (tie stall, free stall or cubicle housing, straw yards). Twenty-two farms with an average of 60 cows in milk per farm (min. 10; max. 220) agreed to enroll in the study. The housing system distribution was the following: 12 farms used free stalls, 7 farms had the cows housed in straw yards, and 3 used permanent tie stalls. All animals were Holstein-Frisian and were milked twice a day. Dry cows (more than 7 months pregnant) were not examined and were not included in the analysis.

In total, 1319 cows in milk were subjected to a simple clinical exam with special emphasis on rib palpation while restrained for routine sanitary interventions (blood sampling and tuberculosis skin testing). When an enlargement was palpated, the rib number was identified by counting the ribs from the 13th forward. The exact location—dorsal/ventral; left/right/bilateral; rib number—of each lesion was registered. The swelling was then pressed and the cow’s response to this was recorded as no reaction, slight reaction, mild reaction, or strong reaction. Reaction was defined as lateral withdrawal movement, rib-cage muscle contraction, or an attempt to kick with the hind limb. 

All animals showing any trace of swelling or enlargement or of pain over any rib were examined in more detail and the clinical history was registered, namely age, last calving date, pregnancy status, current milk yield, history of recent diseases including lameness, hoof lesions, or “downer cow syndrome”, and time to last hoof trimming. A thorough clinical exam was then performed including a hoof examination. Furthermore, lameness was scored on a 1 to 5 scale [17] by making the cow walk on a non-slippery concrete floor. A cow was considered lame if scored 3 or more.

A simplified structured risk assessment of the housing was also completed (Table 1) and the presence or absence of potential risks for rib trauma was recorded, including stall measures, floor quality (e.g., slipperiness), bedding material used, ratio cubicles/cows, ratio cows/feed-trough places, and ease of access to water troughs. In farms with cubicles, a sample of stalls (10 to 20%) was measured in each farm and the average values were used for the analysis. The length was measured from the brisket board to the end curb and the width was measured from one side rail to the other at the level of the neck rail. The height of the brisket board and the height of the neck rail were measured down to the level of the lying surface (considering the bedded area at the time). The end curbs were considered exposed if they were, on average, very evident above the lying surface of the stall.

In cubicles and straw yard farms, the useable area of all water points in the pens was measured and the total length was divided by the number of cows inside the pen. It was considered insufficient if it was below 6 cm per cow. 

No cow had access to pasture, but in some farms, cows had access to a small loafing area. This was considered only in the case of the tie stalls, as in the other systems animals were able to move freely. 

The alleys’ pavement was considered non-slippery if grooves were evident.

Cows with horns were only present in the tie-stall farms. 

Total pen area, to calculate animal density, was not measured in any of the free-stall farms. We also did not look at differences in milk yield between animals with and without rib lesions, although it would be expected that the affected animals would have lower production, taking into account the chronic lameness history.

### 2.2. Statistical Analysis

The measures included in the risk assessment evaluation were used to generate explanatory variables included in univariable linear regression models where the rib lesion prevalence was the outcome variable. Pearson’s Chi-square and Fisher’s exact test were used to evaluate the variables as potential risk factors for the presence of coastal lesions. A *p* value of less than 0.05 was considered statistically significant. All statistical procedures were performed using R software version 4.3.0 [19].

## 3. Results

Of the 22 farms enrolled in the study, 12 did not have any cow showing rib lesions and 10 had at least one animal showing some kind of lesion. The mean prevalence of rib lesions across the study farms was 2.3%, or 31 cows in 1319 evaluated. The range was 0% (12 farms) to 6.1% (1 farm). Affected animals had an average of 3.7 lactations. The distribution of the lesions’ location can be seen in Table 2. No lesion was evidently painful to palpation. 

Table 3 shows cows’ characteristics associated with the occurrence of rib lesions. The close association with lameness is evident with more than 85% of the cows showing some sort of rib lesion with a history of lameness.

The farm risk factors (Table 4) associated with an increase in rib lesions prevalence in the overall model were the presence of too-narrow cubicles (*p* < 0.01), high cow/cubicle ratio (*p* < 0.05), high cow/feed places ratio (*p* < 0.01), an exposed rear curb (*p* < 0.05), and farms that did not include functional hoof trimming in their routine (*p* = 0.05). More than 50% of the cows showing rib lesions were found in the cubicle system. However, it is important to notice that in only three tie-stall farms, there is a considerable amount of affected animals (all tie-stall farms had at least one animal with rib lesions). These differences were not considered significant because of the small numbers of farms using straw yards and particularly using tie stalls. 

All farms had similar and relatively good access to the milking parlor (enough space, no narrow exits, and non-slippery floors), so these risk factors were not analyzed. 

## 4. Discussion

This study shows that rib lesions may be undervalued as a cause of poor welfare in dairy cows and that they may even be relatively common in some Portuguese farms. However, the prevalence of rib lesions in our study (0 to 6.1%) contrasts with the results from the only other published study looking for similar disorders on farms [20]—an overall prevalence of 7.54%, ranging from 0% to 37.49% in a total of 2134 cows assessed in 96 German dairy farms. However, the German study was conducted exclusively on cows kept in tie stalls, which may explain the very high prevalence in some farms. It is well known that tethering impacts animals’ movement, namely when lying down, and this has been associated with a higher prevalence of locomotor disorders [3,15,18,21]. In our study, the tie-stall farms had 14% of all the animals with rib lesions, although they only represent 7% of the total population (92 animals in 1319). For example, increasing chain length in tied animals appears to help in reducing the prevalence of limb injuries by allowing for better lying-down movements [3]. As discussed below, lesions on the more cranial ribs probably result from repeated trauma when cows lie down awkwardly, especially on hard floors. Anyhow, it is noteworthy that some farms do not show any rib lesions, as was the case in our study.

Supporting the hypothesis that some farm characteristics are risk factors for rib lesions, our study showed a strong correlation between housing conditions or poor lameness management and the prevalence of injuries over these bones. We also explored possible correlations between the presence of rib swellings and the animal’s clinical history. By doing so, we suggest that these lesions may be valid indicators of poor housing and severe health-welfare problems (e.g., chronic lameness). Thus, the inclusion of rib lesions as an animal-based indicator in dairy cattle on farms or during slaughter welfare assessment is advocated. 

In terms of individual risk factors, having a history of chronic lameness (27/31 cows) was significantly correlated with showing cranial–ventral lesions (7th to 9th rib). In contrast, ongoing lameness (16/31 cows) or a downer cow history, were not considered significant. The association between chronic lameness and the occurrence of rib lesions was also found in other studies [12,20,22,23,24]. In one of these studies, rib swelling prevalence was 5% in cows with no signs of lameness and 47% in cows with evident signs of lameness [24]. These results are in accordance with what Blowey [5] suggested to be one of the possible origins of this disorder—thin lame cows are more prone to injury. Several factors may explain this correlation: for example, lameness will influence the lying down movement [25,26], namely by hindering a sustained lowering movement ending up with the ribs crashing against the front limb hooves, stall structures, or the floor, especially if bedding is absent or very scarce. In our study, several cows showed bilateral lesions on the cranial ribs showing that this thwarted lying down movement occurs independently of the side of the lame limb. This unsupported collision with the floor is more probable when lying down in stalls, explaining why rib lesions are much less frequent in straw yards [5]. 

Farms that only take cows to chutes to be trimmed when lameness is clinically evident have a much higher lameness incidence [27]. The fact that farms that did not perform functional trimming as a management tool for lameness control showed higher rib-lesion prevalence is probably related to the higher lameness prevalence expected in these farms [28], further affecting the lying down movement and the possibility of ribs bumping against large hooves [25]. Functional trimming (ideally at drying off) is an important way of preventing lameness by maintaining the correct size and shape of hooves [28]. 

Stall characteristics have been proven to be related to the occurrence of different kinds of lesions and poor welfare [1,2,3,29]. Possible causes for the cranial–ventral lesions are the type of stalls (stalls versus straw yards) and the dimensions of the stalls, especially the width, and bedding. A significant correlation between stall width and the occurrence of injuries in both tie stalls and free stalls has been demonstrated [1,29]. Abele et al. [20] found that in some tie-stall farms, over 35% of cows had rib swellings, and a PhD study in the UK [23] looking at 2000 cows from 13 dairy herds reported a prevalence of palpable swellings at the costochondral junctions from the seventh to ninth ribs to be 3.6% to 26.8%. The disparity of these results demonstrates that housing plays a significant role both in tie-stall as well as free-stall systems. Our study also showed a significant correlation between some stall features and the probability of having rib lesions—narrow stalls and an exposed and traumatic curb on the back limit in both free stalls and tie stalls. In contrast, the quality (depth) of the bedding material did not correlate with the prevalence of lesions in our study, although it has been demonstrated that the type and depth of bedding material may influence how the animals lie down [30,31]. Interestingly, rib lesions were not correlated with other limb injuries, such as bursitis or hock lesions, although these may also result from inadequate stalls or poor bedding [32]. 

Another factor significantly affecting the prevalence of lesions was the high cow ratio cubicle per cow (ratio above 1), occurring in 9 of the 12 free stall farms. It is recommended [18] that the number of cubicles should be 5 to 10% more than the number of animals to avoid competition and the need to occupy tight or defective cubicles (e.g., poor bedding, broken structures, etc.), conditions in which lying becomes much more complicated. By having less than enough cubicles, more submissive cows will lie on the alleys or on badly kept stalls or simply remain standing for longer periods, which will increase the possibility of lameness [29,32]. 

Lesions found on the more caudal ribs (12th or 13th) were exclusively dorsal and most certainly resulted from collisions with the dividing rails or lying against them for long periods. When cubicle dimensions are not adequate for the average size of the animals, cows will hit the structures or will lie with their ribs or vertebrae compressed against the metal sides of the stalls [3,29]. Collisions with cubicle structures (e.g., limiting rails) are one of the indicators used in the Welfare Quality protocol [16,32] to assess the Good Housing criteria. These collisions and other traumatic contact with rails were frequently observed and registered in another study in which the Welfare Quality protocol was applied to farms very similar to the present ones [33]. In the present study, caudal–dorsal lesions were only registered on farms with cubicles and were more frequent in those farms that had smaller than what was considered to be ideal spacious cubicles (220–260 cm × 110–120 cm) [3,18,29]. The here-shown correlation between rib lesions and cubicle dimensions supports the possibility of using the prevalence of these lesions as an indicator of stall inadequacy or as an indirect indication of collisions with structures.

Another factor found to be significantly correlated with lesions was the exposure of the rear curb, potentially affecting the lying down movement and comfort. The Welfare Quality welfare assessment protocol [16] uses the indicator “Animals lying partly or completely outside the lying area”, which is a way of assessing comfort around resting by identifying cows lying over or pressing against the curb. The purpose of the curb is to reduce bedding material waste, and ideally, it should be at the level of the lying surface and not above it. In our study, stalls with an exposed or salient traumatic curb (sharp edges versus rounded edges) were associated with more lesions probably because cows reduce the time lying down in these uncomfortable conditions or because cows try to adopt alternative but awkward positions to avoid lying down on the preeminent back edge of the stall [1,2,3,29,32]. A study looking at stall length found that long stalls were associated with more time lying and longer lying bouts than cows in short stalls [1]. This suggests that short lying space will lead to animals standing for longer durations (increasing lameness prevalence) but also adopting more uncomfortable positions, increasing the probability of hitting the stall divisions. 

The high cow/feed spaces ratio (above 1) seen in some of the free stall and straw-yard farms, was also found to be correlated with higher rib lesions’ prevalence. It can be speculated that because of lack of space, head butts between animals competing to reach the feed would be more frequent [34]. Unfortunately, total stock density was not calculated in the free stall farms, as overstocking could be associated with agonistic behaviors and namely with head butts. However, these head butts would probably lead to a very different kind of lesion (large, edematous, and painful) as it is implausible that multiple hits on the same area would occur. Also, there was no significant correlation between herds that kept some animals with horns and the presence of rib lesions, which would be expected if head-butts were responsible for the lesions. Thus, it is more probable that this shortage of feeding spaces occurs in farms with many other housing and management problems, which are the true risk factors for rib lesions. For example, it has been shown that one of the herd-level risk factors for lameness is overcrowding [35,36], suggesting that it could also be a risk factor for rib lesions, as explained above.

Whatever the source or the rib affected, it is not conceivable that one hit, fall, or head-butt would cause this type of hard non-painful rib lesions. It is much more likely that multiple, frequent traumas will cause tissue (bone) reactions leading to the formation of fibrosis and finally exostosis [11,12,13,22]. The chronicity and slow evolution would explain the lack of pain induced by palpation that was found in the clinical exams of our cows. However, even if not a source of acute and constant pain, these lesions reflect the prevalence of lame animals and poor husbandry conditions, and so could be used as a welfare indicator in dairy farms. 

## 5. Conclusions

This study confirms that rib lesions caused by trauma are relatively frequent in some dairy farms while being completely absent in others. Our results show that cows that have a clinical history of lameness associated with problems in lying down gently and/or having to lie down in stalls that thwart their lying movements have an increased probability of suffering small but constant hits to their ribs. These repeated and continuous traumas result in evident and palpable lesions over certain ribs. However, we did not find the high prevalence that was depicted in other studies, probably because of lower lameness incidence or more spacious stalls in our farms, supporting the idea that these differences result from specific housing and management shortfalls occurring in some farms, especially affecting cows’ lying behavior and lameness prevalence.

Although our results are useful in providing evidence that rib lesions may be prevented by good husbandry practices (adequate cubicle dimensions, suitable animal density, and establishment of lameness prevention measures), there were some limitations to the analysis performed. For example, it was not possible to prove a causative relationship between the individual clinical variables measured and an increased risk of rib lesions because we did not record the prevalence of cows with no rib lesions showing the same clinical conditions. These limitations should be considered when interpreting the results, as it is possible, although not plausible, that other cows also exposed to these inadequate stalls and probably with a history of lameness do not develop these lesions. Other factors such as weight and size of the animals, udder size, pregnancy status, and milk yield, could also play a role.

## Figures and Tables

**Table 1 animals-14-00338-t001:** Risk factors associated with housing for rib lesions with an indication of the thresholds considered acceptable. Thresholds are based on recommendations on EFSA 2023 Scientific Opinion [18].

Risk Factor	Thresholds for Acceptability	Criteria
Ratio cows/cubicle	≤1	At least one stall per cow in barns with cubicles
Ratio cows/feed places	≤1	At least one place per cow (free stall and straw yards)
Access area to water troughs	>6 cm	Linear water troughs access area in cm per cow
Loafing area	Yes	Access to an exercise area at least 1 h per day
Stall length	≥220 cm	The average of a sample measured in each farm
Stall width	≥110 cm	The average of a sample measured in each farm
Brisket board height	≥10 cm	Ensures that cows do not lean forward too much
Bedding (quality)	Good	The concrete under the bedding is not visible
Neck rail distance from the curb	>1.73 m	Indicates easiness in getting up
Alley floor	Not slippery	Presence of evident grooves.
Animals with horns	No	No cows with horns in the free stall systems
Exposed rear curb	No	Curb is not visible above the bedding material

**Table 2 animals-14-00338-t002:** Prevalence of lesions in dairy cows according to rib number.

Rib or Ribs Affected		Prevalence
Unilateral	7th	3%
	8th	23%
	9th	11%
	10th	3%
	11th	5%
	12th	6%
	13th	20%
Bilateral	8th	26%
	9th	3%

Note: cows may have more than one rib affected.

**Table 3 animals-14-00338-t003:** Animal risk factors based on clinical examination and clinical history.

Clinical History and Clinical Signs	Cows Affected Showing Rib Lesions (%)
Downer cow (during the previous year)	16.13
Lameness at any time during previous year	87.1
Recent claw disease (treated)	11.11
On-going claw disease	51.61
Bursitis forelimbs	12.90
Bursitis hind limbs	3.23
No functional claw trimming	51.61
Sporadic claw trimming	32.25
Routine claw trimming	16.12
Not currently lame	48.38
Currently lame	51.61

**Table 4 animals-14-00338-t004:** Farm risk factors and percentage of cows with rib lesions that were exposed to them.

Farm Risk Factors		N° Farms	Cows Showing Rib Lesions Exposed to Risk Factor (%)	*p* Value
Type of housing	Free-stall housing (cubicles)	12	54.54	
	Straw-yard housing	7	31.81	
	Tie-stall housing	3	13.63	
Claw trimming frequency	Routine trimming	7	31.81	0.05
	Only when lame	15	68.18	
Flooring	Acceptable	9	40.90	
	Slippery	13	59.09	
Ratio cow/cubicle	>1	9	75.0	<0.05
	≤1	3	15.0	
Ratio cow/feed space	>1	13	68.42	<0.01
	≤1	6	31.58	
Access to water trough	Easy	10	45.45	
	Difficult	12	54.54	
Loafing area	Yes	9	40.90	
	No	13	59.1	
Stall measures	Adequate (≥220 length; ≥110 cm width)	4	33.33	<0.01
	Inadequate	8	66.66	
Stall bedding	Good bedding	10	16.67	
	Inadequate bedding	2	83.33	
Bedding in other	Good bedding	8	20	
	Bad bedding	2	80	
Brisket board	Yes	8	75	
	No	2	25	
Rear curb	Not exposed	2	16.66	<0.05
	Exposed	10	83.33	
Maternity	Yes	7	31.81	
	No	15	68.18	

## Data Availability

The data presented in this study are available upon request from the corresponding author.

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
