# Peer review of "Prevalence and Risk Factors Associated with Rib Lesions in Dairy Cows"

_animals, 2024, doi:10.3390/ani14020338_

Round 1
Reviewer 1 Report
Comments and Suggestions for Authors
The topic of the manuscript is very relevant and of practical value.
I have the following notes and recommendations for the authors:
1. The name of the tables should be placed above the table as per the requirements of the journal.
2. In the material and method section, note whether all cows in the farms included in the study are hornless because it is particularly important for the presence of animal trauma. To describe the average area available to each free-range cow, because it is very important for hierarchical relationships between animals and cases of trauma.
3. In the case of pregnant cows, it should be noted how pregnant the studied cows are, because a pregnancy of up to 3-4 months should not affect the behavior when lying down, as it would affect at 7 months, for example. If the studied cows are of different pregnancies, they should be divided into at least three groups, which is much more objective when considering the influence of this factor.
4. To describe the daily milk yield and if there are differences of more than 10 kg per day, to divide the farms according to this indicator as well.
5. To look for the dependence of the injuries with the milking system and more precisely the pushers that bring the cows into the waiting room and after it. Very often the exits or entrances during milking are narrow or with very sharp turns, in which cows are pressed, fall and complications and injuries occur on various parts of the body.
Author Response
The topic of the manuscript is very relevant and of practical value.
I have the following notes and recommendations for the authors:
A: We thank the reviewer for these good suggestions that have improved the quality of our paper.
- The name of the tables should be placed above the table as per the requirements of the journal.
A: Corrected.
- In the material and method section, note whether all cows in the farms included in the study are hornless because it is particularly important for the presence of animal trauma.
A: All cows in the free stalls (cubicles and straw yards) were hornless, but not in the tie-stall. This information has been added.
To describe the average area available to each free-range cow, because it is very important for hierarchical relationships between animals and cases of trauma.
A: We agree that this is an important characteristic but unfortunately we did not measure the pens’ area. We have added this information in the Material and Methods and included a comment in the discussion (Line 264).
- In the case of pregnant cows, it should be noted how pregnant the studied cows are, because a pregnancy of up to 3-4 months should not affect the behavior when lying down, as it would affect at 7 months, for example. If the studied cows are of different pregnancies, they should be divided into at least three groups, which is much more objective when considering the influence of this factor.
A: This is true – heavy pregnant cows will lie down differently. Regrettably we did not register the pregnancy status of the affected cows because we only examined in-milk cows (see Line 109). So animals with pregnancy above 7 months were not included in the calculations.
- To describe the daily milk yield and if there are differences of more than 10 kg per day, to divide the farms according to this indicator as well.
A: Again a very perceptive comment. Again, this data was not registered so analysis taking into account total milk yield is not be possible. We also suspect that if we divided the group up we would have difficulty in getting some statistically significant results. Also we suspect that the cows with rib lesions would be in the low quartile as these were mostly cows with a history of chronic lameness. We included a few sentences on this in the Conclusions (Line 297).
- To look for the dependence of the injuries with the milking system and more precisely the pushers that bring the cows into the waiting room and after it. Very often the exits or entrances during milking are narrow or with very sharp turns, in which cows are pressed, fall and complications and injuries occur on various parts of the body.
A: Milking parlour features are also very important and we did look at them. However, there were no evident differences between farms and we did not find any important flaw in the structure or pavement. Also, no farm had a pusher system. In Line 160 we state this “All farms had similar and relatively good access to the milking parlour (enough space, no narrow exits and non-slippery floor), so these risk factors were not analysed”.
Reviewer 2 Report
Comments and Suggestions for Authors
The authors Stilwell et al., investigated the potential risk factors linked with the rib lesions in dairy cows. The manuscript is written clearly with several additional references. Since the article needs improvement in the methods and results sections. The comments have been given below.
Major comments
1. The authors should include additional information about Table 1 in the methodology section.
2. The results section should update on the prevalence of rib lesions and the reasons for the rib lesions related to farm risk factors.
3. Line 181-190: Do the authors have any evidence of photos of rib swelling from this study? Please provide them in the supplementary files.
Minor comments
1. Line 22 and 68: Rewrite sentence.
2. Line 165. The author informed about the Portuguese farms, but again discussed about the German dairy farm in line 168.
3. Line 178: What are the environmental risk factor involved in the tied cow farms?
4. Line 255: Reference bracket.
5. Line 293: The authors must present an interpretation of the results, which is unclear.
Minor editing of English language required.
Author Response
The authors Stilwell et al., investigated the potential risk factors linked with the rib lesions in dairy cows. The manuscript is written clearly with several additional references. Since the article needs improvement in the methods and results sections. The comments are given below.
Major comments
- The authors should include additional information about Table 1 in the methodology section.
A: We thank the reviewer for the suggestion. We do agree that the Table is not self-explanatory for those not used to evaluate these features.
- The results section should update on the prevalence of rib lesions and the reasons for the rib lesions related to farm risk factors.
A: We tried to discuss the rib lesions correlations with the farm risk factors in the Discussion. We are not sure what should be updated in the results section, but we tried to make it more clear by adding a few explanation sentences.
- Line 181-190: Do the authors have any evidence of photos of rib swelling from this study? Please provide them in the supplementary files.
A: Yes, we have a few, although the lesions are not very clear because some are difficult to visualize but are very evident by palpation. If the reviewer and Editor wish, we can add them as supplement material.
Minor comments
- Line 22 and 68: Rewrite sentence.
A: Sentences rewritten.
- Line 165. The author informed about the Portuguese farms, but again discussed about the German dairy farm in line 168.
A: Not sure what is suggested. Is there a contradictory? In these lines we compare our results with other that also found a strong correlation between lameness and the occurrence of the lesions. Could the reviewer please advise on what should be added?
- Line 178: What are the environmental risk factor involved in the tied cow farms?
A: In the tie stall we also may have an exposed curb and tight lying area leading to collisions with the rails. But the risk factor usually related to this system is the difficulty in lying down easily and the high lameness prevalence in some farms. We have added a few sentences explaining these possibilities.
- Line 255: Reference bracket.
A: Amended.
- Line 293: The authors must present an interpretation of the results, which is unclear.
A: We agree and thank the reviewer for the insight. We have added a few sentences showing what could improve the evidence of the results, although we also reinforce the idea that our results together with those of similar studies, do show a strong correlation between lameness and inadequate structures and rib lesions.
Round 2
Reviewer 2 Report
Comments and Suggestions for Authors
I appreciate the authors for all the modifications and responses given to all the questions raised by the reviewers. I have clarified the questions that the authors were unclear on.
Q. We are not sure what should be updated in the results section, but we tried to make it more clear by adding a few explanation sentences.
A. I requested explanations for the results in the results sections. Previously, a simple paragraph was not very clear about the tables and their results.
Q: Yes, we have a few, although the lesions are not very clear because some are difficult to visualize but are very evident by palpation. If the reviewer and Editor wish, we can add them as supplement material.
A.I have not yet been able to access the supplementary figure, which is not provided with the manuscript and is not available on the site (I see the description). This figure may support the authors' observations of rib lesions (scar tissue area near the lesions, swelling indications, and surface hair loss) in dairy cows.
Q: Not sure what is suggested. Is there a contradictory? In these lines we compare our results with other that also found a strong correlation between lameness and the occurrence of the lesions. Could the reviewer please advise on what should be added?
A. Now, it is clear with the revised version.
I request the authors to follow same name, for ex. Lisbon in some places and "Lisboa" and Line 109 for 1,319 change to 1319 similar to others.
Result part_Line 166: No lesion was evidently painful to palpation? Have the authors used or measured any parameters for the pain during assessment ?
Author Response
I appreciate the authors for all the modifications and responses given to all the questions raised by the reviewers. I have clarified the questions that the authors were unclear on.
A: We again thank the Reviewer for the suggestions and clarifications. We hope that now we have answered everything.
- We are not sure what should be updated in the results section, but we tried to make it more clear by adding a few explanation sentences.
- I requested explanations for the results in the results sections. Previously, a simple paragraph was not very clear about the tables and their results.
A: We think that the explanation is now more comprehensive.
Q: Yes, we have a few, although the lesions are not very clear because some are difficult to visualize but are very evident by palpation. If the reviewer and Editor wish, we can add them as supplement material.
A.I have not yet been able to access the supplementary figure, which is not provided with the manuscript and is not available on the site (I see the description). This figure may support the authors' observations of rib lesions (scar tissue area near the lesions, swelling indications, and surface hair loss) in dairy cows.
A: We have added a photograph that we think illustrates the type of lesion. Generally, there is no scar, no loss of hair or any other external sign except for the tumefaction.
Q: Not sure what is suggested. Is there a contradictory? In these lines we compare our results with other that also found a strong correlation between lameness and the occurrence of the lesions. Could the reviewer please advise on what should be added?
- Now, it is clear with the revised version.
A: We again thank for the request for a clearer conclusion.
I request the authors to follow same name, for ex. Lisbon in some places and "Lisboa" and Line 109 for 1,319 change to 1319 similar to others.
A: Amended, as suggested.
Result part_Line 166: No lesion was evidently painful to palpation? Have the authors used or measured any parameters for the pain during assessment?
A: This is a good point. Cows tend to disguise pain, so only subtle reactions were expected. The ones we looked for are now described in the text.